# Prevalence of undernutrition and associated factors among street adolescents in adama town, oromia regional state, Ethiopia, 2023: A cross sectional study

**Tsinukal Tesfaye[1], Ebissa Bayana Kebede[2], Vinod Bagilkar[2], Fentahun Meseret[3]** *

**1** Department of Pediatrics and Child Health Nursing, College of Health and Medical Science, School of Nursing and Midwifery, Diredawa University, Diredaw, Ethiopia, **2** Faculty of health sciences, Department of Pediatrics and Child Health Nursing, Institute of Health, School of Nursing, Jimma University, Jimma, Ethiopia, **3** Department of Pediatrics and Child Health Nursing, College of Health and Medical Science, School of Nursing and Midwifery, Haramaya University, Harar, Ethiopia

* mesie1055@gmail.com

**Data Availability Statement:** All relevant data are within the paper.

## Abstract

### Background

Undernutrition remains a serious public health problem in developing countries, including Ethiopia. In particular, street adolescents are more at risk for undernutrition because they are the most underprivileged population. However, there is a paucity of information about undernutrition among street adolescents in Ethiopia.

### Objective

To assess the prevalence of undernutrition (stunting and thinness) and factors associated with undernutrition among street adolescents in Adama town, Oromia regional state, Ethiopia, 2023.

### Method

A community-based cross sectional study was conducted among 358 street adolescents from January 15–30, 2023. A convenience sampling technique was used to select the required sample size. A structured, interviewer-administered questionnaire was used to collect the data from the respondents. The collected data were checked, coded, entered into Epidata 4.6 and exported to Statistical Package for Social Sciences version 25. Both Bivariable and Multivariable logistic regression analyses were conducted to determine the factors that are associated with thinness and stunting. P values below 0.05 at the 95% confidence interval were considered indicative of a statistically significant association. Finally, statements, graphs, tables and charts were used for result presentation.

### Result

This study revealed that, the prevalence of undernutrition was 47.2%. Thinness and stunting accounted for 20.4% and 34.1% respectively and 7.3% both thinness and stuting. Age

**Funding:** The author(s) received no specific funding for this work.

**Competing interests:** The authors have declared that no competing interests exist.

**Abbreviations:** AOR, **A**djusted **O**dd **R**atio; BMI, **B**ody **M**ass **I**ndex; CDC, **C**ommunicable **D**isease **C**ontrol; CI, **C**onfidence **I**nterval; COR, **C**rude **O**dd **R**atio; DDS, **D**iatery **D**iversity **S**core; HFA, **H**eight **F**or **A**ge; IRB, **I**nstitutional **R**eview **B**oard; NNP, **N**ational **N**utrition **P**rogram; PHQ, **P**atient **H**ealth **Q**uestionnaries; SD, **S**tandard **D**eviation; SPSS, **S**tatistical **P**ackage **F**or **T**he **S**ocial **S**ciences; UNICEF, **U**nited **N**ations **I**nternational **C**hildren's **E**mergency **F**und; WHO, **W**orld **H**ealth **O**rganization; WFA, **W**eight **F**or **A**ge.

(adusted odd ratio = 1.41; 95% confidence interval: 1.17–1.71), skipped one or more meals per day ((adusted odd ratio = 3.50; 95% confidence interval: 1.23–9.94), drinking unprotected water source ((adusted odd ratio = 3.23; 95% confidence interval: 1.49–6.98) and use of mastish ((adusted odd ratio = 2.91; 95% confidence interval: 1.19–7.12) were factors statistically associated with thinness. Being skipped one or more meals per day ((adusted odd ratio = 4.14; 95% confidence interval: 1.87–9.14), washing hands before meals ((adusted odd ratio = 0.46; 95% confidence interval: 0.26–0.81) and moderate depression ((adusted odd ratio = 2.93; 95% confidence interval: 1.05–8.15) were factors significantly associated with stunting.

## Conclusion and recommendation

In conculusion, the prevalence of undernutrition (thinness, stunting or both together) was high among street adolescents. To enhance street adolecents' nutritional status, targeted nutritional treatments, providing health services and good hygiene and sanitatios practices are urgently needed.

## Introduction

Undernutrition is a consequence of a deficiency in nutrient intake and/or absorption in the body or It can result from a disease or infection that affects nutritional intake or metabolism [1,2]. According to the World Health Organization (WHO), undernutrition includes wasting, stunting, underweight, and micronutrient deficiencies [2]. Undernutrition is the main nutritional problem affecting adolescent populations worldwide indicated by stunting and thinness [3]. It is assessed by using appropriate anthropometry, rapid dietary assessment, physical examination, laboratory assessment, and household food security [2,3]. Stunting, also known as chronic undernutrition, is presents as low height-for-age (HFA). It is a result of prolonged or repeated episodes of undernutrition and is defined by HFA below minus two standard deviations (<-2SD) from the median height for age of the reference population [2].Where as, thinness is defined by a body mass index (BMI) for-age below minus two standard deviations (<-2SD) from the reference population [3,4] and It is a sign of poor nutritional conditions and indicates decreased fat and muscle mass;BMI is therefore, helpful in determining how much of a weight deficit there is in relation to height among adoulescent population [3–6].

According to WHO and The Ethiopian National Nutrition Program (NNP) definition, adolescence is an important and transitional period between childhood and adulthood, which categorizes the age range as 10–19 years [3,7]. During this period, there is rapid face of growth and development with a great need ofadequate nutrition to sustain optimal groth and development [6,8,9]. Currently, there are nearly 1.2 billion adolescents worldwide [6]. Over 90% of adolescents live in sub-Saharan Africa and Asia [10]. In Ethiopia, they form 33.8% of the total population [11].

Globally, undernutrition is the primary risk factor that increases the morbidity and mortality of street adolescents [5]. According to the United Nations International Children's Emergency Fund (UNICEF), there are two types of street children: those who live and sleep in public areas ("off the street") and those who own homes but spend the day on the street ("on the street") [12]. It is estimated that, there are over 150 million worldwide [13]. Around 10

million are thought to exist in Africa [13]. In Ethiopia, there are around 600,000 street children living in the country [11].

Street adolescents experience a wide range of nutritional problems as a result of their unhealthy lifestyles and challenging environmental settings including lack of access to basic health and nutritional care services [14]. Globally, undernutrition is the primary risk factor that increases the morbidity and mortality of street adolescents [5,14], approximately 80 million street children suffer from stunting and wasting [15]. In low and middle income countries, such as Asia (26%) and Africa (45%), adolescent deaths occur due to undernutrition [2]. In India and, Dhaka53% and 73% of street children suffer from chronic undernutrition [16,17] respectively. Which signifys that, the problem of undernutrition in street adolescents is serious in low and middle income countries in which many resources are limited. Similarly, in some African countries, there is a high rate of undernutrition among street children [18–20]; In Ethiopia, a study done in Jimma, thinness and stunting acounts 29.2% and 30.4%, respectively [21]. In Northwest Ethiopia (Bahirdar and Gondar) wasting and stunting also accounts 15.3% and 46.4%, respectively [18].

The risk factors for undernutrition were blamed to be due to their vulurability; consume few unhealthy nutritious foods, have limited access to clean water, lack access to shelter and hygienic practices, lack information and health care, and live in unhealthy surroundings [15,16,22]. Being addicted to psychoactive drugs also causes undernourishment through decreased intake, nutrient absorption, and dysregulation of hormones that alter the mechanisms of food intake [23]. Moreover, street adolescents' undernutrition and failure to thrive are exacerbated by societal discrimination, emotional abuse, a lack of strong parent-child bonds, sadness, fear, hopelessness, helplessness, suicidal ideation, and loneliness, which lead to severe mental health problems, especially depression and anxiety [22] which inturn ultered their overall development including sexual maturation and an increased risk of infection and early mortality [9,22,24–26].

As result of this, undernutrition can maintains poverty, which further prevents economic and social development [26–28].

In Ethiopia, government offices and different civil organizations are working in collaboration in different cities to overcome the problems that, street adolescents faces [29]. To assist street children and reduce the number of children living on the street, the Social Service Society Children's Village (SOS) has also started a project in the three major cities in Ethiopia; namely, Addis Abeba, Adama, and DireDawa [30]. The WHO and the Federal Democratic Republic of Ethiopia's Ministry of Health have also developed guidelines about adolescents' nutrition to promote a healthy diet [2,3]. Our intension here is therefore, to highlight and show the outcomes of the effort made by these organization in Adama town and to point out additional remedial actions that must be carried out by the responsible bodies based on the current findings.

Despite the efforts made by the government and other stakeholders mentioned above, undernutrition among street adolescents remain a public health problem in Ethiopia [31]. The current situations in the country such as political instability, poverty, drought, and the high cost of living in the country aggravates the problem of undernutrition among street adolescents [32]. We also noted that, they are most neglected population who didn't get an attention or permanent solutions to their problems from the public health sectors, academicians, policymakers, or the government [11]. Moreover,There has not been enough research, particularly in Adama town, where there are many street adolescents are concentrated. Therefore, this research is aimed to assessing the prevalence of undernutrition and its associated factors among street adolescents in Adama, Oromia Regional State, Ethiopia.

## Method

### Study setting, design and period

A community based cross sectional study was conducted in Adama town, which is located in the Oromia Region state of Ethiopia and surrounded by the East Shawa Zone. It is located approximately 99 km southeast of Addis Ababa, the capital city of Ethiopia. It is the third largest urban center and the main city where multiple migrations, including internal migration from conflict, and trafficking, occur. Administratively, Adama town is divided into 14 urban kebeles and, according to the projected population data in 2019, the size of the population of the city of Adama was 433,046 of which 216,046 and 217,000 were male and female, respectively. The Bureau of Women's, Children's, and Youth's Affairs in Oromia collaborated with UNICEF to count the number of street children in Adama town, and stated that almost 5000 street children lived in Adama town [33]. The study was conducted from January 15–30, 2023.

### Population

**Source population.** The source populations were all street adolescents whose ages ranged from 10 to 19 years who lived in Adama town, Oromia regional state, Ethiopia.

**Study population.** The study populations were the selected street adolescents who lived in Adama town during the study period.

**Sample size determination.** The sample size was calculated by using the single population proportion formula, where the 95% confidence interval (Z/2), 5% marginal error (d), and 30.4% prevalence of stunting(P) were taken from a previous similar study that was conducted in Jimma, Ethiopia [21]. By considering a 10% nonresponse rate the final sample size became **358.**

**Sampling procedure and technique.** Based on observation and discussions with key informants working in the field of street children, such as the Adama branch of SOS Children's Villages and the Adama City Administration's Labor and Social Affairs Office, we selected five of the 14 kebeles in Adama town that had a high number of street children. Badhatu Kebele, Gada Kebele, Gurmu Kebele, Bokku Shanan Kebele, and Hangatu Kebele were the kebeles that were chosen. Then, a convenience sampling technique was used to select study participants.

**Inclusion criteria.** Street adolescents whose ages ranged from 10 to 19 years who lived on the street or off the street.

**Exclusion criteria.** Street adolescents who were unable to give response due to medical and psycho social problems.

### Variables

**Dependent variable.** Prevalence of undernutrition (either stunting or thinness or both together)

**Independent variable.** Sociodemographic factors (age, sex, educational level, parental education and source of income); Lifestyle related factors (use of substance, place of sleep at night, type of street, number of years on street); Illness and health care service (respiratory infection, diarrhea, GI infection, skin infection, loss of appetite, health care services); Hygiene and sanitation practice related factors (source of water, use of latrine, wash hand before meal, bath taking); Dietary intake related factors (source of food, go to sleep hungry, skipped one or more meals per day, number of times eat per day, Dietary diversity score); Screening for depression related factors (hopeless, less interest, trouble sleeping, weight loss or gain, feeling tired, feeling bad, trouble concentrating, restless).

## Operational definition

**Undernutrition:** It includes either stunting or thinness or both together; **Stunting:** Adolescents who are found to be below -2 SD of HFA and gender; **Thinness:** Adolescents who are found to be below -2 SD of BMI for age and gender [3].

**Adolescent:** A period of transition between childhood and adulthood, whose age is between 10 to 19 [6].

**Street children:** children who live and sleep in public areas ("off the street") and those who own homes but spend the day on the street and spend the night at their homes("on the street") [12].

**Adequate dietary diversity score:** total calculated dietary diversity score (DDS) $\geq$ 5 [34].

**Inadequate dietary diversity score:** total calculated dietary diversity score (DDS) <5 [34].

**Classification of Depression in adolescents:** None 0–4, Mild 5–9, Moderate 10–14, Moderately severe 15–19, and Severe depression 20–27 [35].

## Data collection

Data were collected using a combination of a structured interviewer-administered questionnaire and anthropometric measurements such as height and weight, which were adapted after reviewing different literatures and guidelines [1,18,21,23,25]. The data collection tool included sociodemographic and lifestyle characteristics and substance use, illness and health care, hygiene and sanitation practice, dietary intake practice, screening for depression, and anthropometric measurements. Dietary diversity score and screening for depression were assessed by the tools that were adopted from the Food and Agriculture Organization (FAO) and the Patient Health Questionnaire-9 (PHQ-9) [34,35].

The data were collected by three BSC nurses and supervised by one MSc holder who had previous experience in data collection and supervision. Additionally, we received one employee from the Adama City Administration's Labor and Social Affairs Office to facilitate the data collection with street adolescents.

The nutritional status of the study participants was assessed based on anthropometric measurements. Weight was measured by a PH-2015A portable digital weighing scale for all participants once while wearing lightweight clothing and barefoot. Calibration was performed before weighing each participant by setting it to zero on a digital weight scale with a precision of 0.l kg. The height of each study participant was collected by measuring their height with a tape measure (the tape was carefully attached with a stick to prevent stretching) in a vertical position by having them stand barefoot on a flat surface, with the adolescent's heel, buttock, and occiput making contact with the wall while standing with a precision of 0.1 cm.

According to WHO growth reference data for 5–19 years, undernutrition in adolescents was assessed using the standard indicators of BMI-for-age and Height-for age. BMI was calculated by weight in kg divided by height squared in meter. Those who had a BMI for age less than -2 Z score were categorized as thin, and those who had a HFA Z score of less than-2 Z score were categorized as stunted [36].

The dietary intake of the study participants was assessed by the tool's dietary diversity score in a 24 hour recall period, which was prepared by the FAO. The food groups were categorized into nine food groups and calculated by summing the number of food groups consumed by the individual respondent over the 24-hour recall period. If the individual consumed fewer than five food groups, it was categorized as having inadequate dietary diversity, and if the individual consumed five or more food groups, it was categorized as having adequate dietary diversity [34].

Screening for depression in adolescents was measured by the tools depression for adolescents tool (adapted from PHQ-9 modified for adolescents Patient Health Questionnaries (PHQ)) [35].

## Data quality control

The questionnaires were originally prepared in English and then translated to local languages Afaan Oromoo and Amharic for easy management, then translated back to English to maintain the quality and consistency of the information. To assure the quality of the data, prior to the actual data collection time, the questionnaire was pretested on 18(5%) of the total sample size at Mojo Town. After that, some amendment was done on the questionnaires. The reliability of the overall questionnaires was checked by Cronbach's α and showed 0.66. The validity and reliability test for tools to screen for depression and dietary diversity score was checked by other studies conducted in Ethiopia and found sensitivity(88%), specificity(78%) and Cronbach's α = 0.78 for depression and sensitivity(88%), specificity(71%) and Cronbach's α = 0.76 for dietary diversity score [37,38].

To create a common understanding of the procedures and approaches used during the data collection, a one day orientation was given for a supervisor and three data collectors. The orientation included describing the objectives of the study, variable definitions, ethical issues, confidentiality of information, the general techniques and methods for anthropometric body measurements and dietary assessment. The data were checked for completeness throughout the data collection process by the supervisor and data collectors. All collected data were examined for completeness and consistency during data management, storage, and analysis. To avoid the reselection or reinterview of respondents, the selected participants were colored on their left thumb by Iodin.

## Data processing and analysis

The collected data were checked, coded, entered into Epidata 4.6 and exported to SPSS version 25 for further analysis. Bivariable and multivariable logistic regression analyses were used to determine the effect of factors on the outcome variable representing undernutrition (stunting and thinness) and to control possible confounders. A bivariable analysis was conducted for stunting and thinness separately, and those variables with a P value of $< 0.25$ on the bivariable analysis were entered in the multivariable analysis. Collinearity was checked using standard error. The Hosmer-Lemshow and Omnibus tests were performed to test the model goodness of fit. Then, associations were shown using the adjusted odds ratio (AOR) at a 95% confidence interval (CI), and P values $< 0.05$ were considered statistically significant.

## Results

### Sociodemographic characteristics of study participants

A total of 358 street adolescents were registered in the study. The mean age of the study participants was 14.92(±2.351) years. The majority (212, 59.2%) of them were between 15 and 19 years old. According to the findings, the majority (282, 78.8%) of study participants were male. Concerning place of birth, almost three-quarters (266, 74.3%) of study participants were born outside of Adama. Most of (312, 87.2%) of the study participants lived without their parents or lived alone, and of them, 121(38.8%) mentioned the reason for leaving their parents were to earn money (employment).

Regarding their biological parents, more than half (219, 61.2%) of the study participants mentioned that, their biological parents (both father and mother) were alive.On about

occupational status of their parents, more than half (191, 67.7%) of the mothers of study participants were housewives, and less than 88(35.5%) of the fathers were farmers. Moreover, regarding parental education, 140(49.6%) of the mothers of the study participants were uneducated, whereas 117(47.2%) of the study participants fathers had completed primary school.

On the educational status of the study participants, almost three-quarters (271, 75.7%) completed elementary school and 64(17.9%) were currently attending school.

In regard to their sources of income, less than half (147, 41.1%) of them received their income by begging (Table 1).

## Lifestyle and substance use-related characteristics of study participants

More than half (235, 65.6%) of study participants, slept at night in the street (off street), and from them, the majority (145, 61.7%), slept in beranda. In most of the study participants, 287 (80.2%) the total duration of being on the street was between 6 months and 60 months (5 years).

In terms of substance use, less than half (145, 40.5%) of study participants, used a mastish (sniffing glue) (Table 2).

## Characteristics of the study participants related to illness, health care services, hygiene, and sanitation practices

Regarding their health status, more than half (208, 58.1%) of the study participants had a history of being sick in the past 3 months. From study participants who had sickness 105(50.5%) had respiratory infections. In the case of medical care, 142(39.7%) study participants sought medical care during sickness, of whom 83(58.5%) went to a district hospital. The majority, 102 (71.8%) of the study participants who did seek medical care and went to a health center, had taken the prescribed medication properly.

In the case of hygiene and sanitation practices, 145(40.5%) of study participants used latrine, and 121(33.8%) practiced body bathing per week. This finding also showed that, 212 (59.2%) of them washed their hands before meals, and 279 (77.9%) of the study participants obtained water from a protected source (Table 3).

## Dietary intake practice related characteristics of study participants

This study found that, the majority (297, 83%), of the study participants received their food from hotels, which was leftover. For thie eating frequency, 187(52.2%) of study participants ate twice per day and 272 (76%) of the study participants skipped one or more meals per day; moreover, (260, 72.6%) of the study participants had gone to sleep hungry.

On their dietary diversity score, 281(78.5%) of the study participants had adequate daily dietary diversity scores (Table 4).

## Prevalence of undernutrition, stunting and thinness among street adolescents

The mean weight was 41.05 (±7.98 SD) ranging from 20.9–66.2 kg and the mean height was 152 cm (±0.11 SD) ranging from 124–175 cm.

The overall prevalence of undernutrition was 169(47.2% (95% CI: 41.9%-52.0%)). From this stunting and thinness accouted 122(34.1%), and 73(20.4%) respectively and 26(7.3%) both stunting and thinness. (Figs 1 and 2).

**Table 1. Sociodemographic characteristics of street adolescents and their parents in Adama Town, Oromia regional state, Ethiopia, 2023 (N = 358).**

| Characteristics | Category | Frequency | % |
|---|---|---|---|
| Sex | Male | 282 | 78.8 |
| | Female | 76 | 21.2 |
| Age | 10–14 | 146 | 40.8 |
| | 15–19 | 212 | 59.2 |
| Place of birth | Adama | 92 | 25.7 |
| | Other than Adama | 266 | 74.3 |
| Living with parents | No | 312 | 87.2 |
| | Yes | 46 | 12.8 |
| Reason for leaving your parents(n = 312) | Death of parents | 52 | 16.7 |
| | Family breakdown/divorce | 24 | 7.7 |
| | Poverty | 72 | 23.1 |
| | To earn money/employment | 121 | 38.8 |
| | Peers pressure | 31 | 9.9 |
| | Abuse | 8 | 2.6 |
| | Others | 4 | 1.3 |
| Natural parents alive | Both alive | 219 | 61.2 |
| | Both dead | 46 | 12.8 |
| | Mother only dead | 29 | 8.1 |
| | Father only dead | 63 | 17.6 |
| | Didn't know | 1 | 0.3 |
| Mothers job (n = 282) | Housewife | 191 | 67.7 |
| | Farmer | 18 | 6.4 |
| | Merchant | 43 | 15.2 |
| | Employment(governmental or nongovernmental) | 3 | 1.1 |
| | Private job | 3 | 1.1 |
| | Daily worker | 16 | 5.7 |
| | Other | 8 | 2.8 |
| Fathers job (n = 248) | Unemployed | 46 | 18.5 |
| | Farmer | 88 | 35.5 |
| | Merchant | 38 | 15.3 |
| | Employment(governmental or nongovernmental) | 14 | 5.6 |
| | Private job | 28 | 11.3 |
| | Daily worker | 26 | 10.5 |
| | Other | 8 | 3.2 |
| Mothers Educational status (n = 282) | Elementary school | 108 | 38.3 |
| | Secondary school | 27 | 9.6 |
| | Higher education | 7 | 2.5 |
| | No formal education | 140 | 49.6 |
| Fathers educational status (n = 248) | Elementary school | 117 | 47.2 |
| | Secondary school | 49 | 19.8 |
| | Higher education | 6 | 2.4 |
| | No formal education | 76 | 30.6 |
| Educational status of the street adolescents | Elementary school | 271 | 75.7 |
| | Secondary school | 24 | 6.7 |
| | No formal education | 63 | 17.6 |
| Currently attending school | No | 294 | 82.1 |
| | Yes | 64 | 17.9 |
| Source of income | Carrying | 39 | 10.9 |
| | Begging | 147 | 41.1 |
| | Shoeshine | 31 | 8.7 |
| | kurale | 29 | 8.1 |
| | Daily trading | 10 | 2.8 |
| | Petty trading | 38 | 10.6 |
| | Garbage picker | 48 | 13.4 |
| | Others* | 16 | 4.5 |

*Others sources of income include: Weight scaling and car washing.

**Table 2. Life style and substance use-related characteristics of street adolescents in Adama town, Oromia regional state, Ethiopia, 2023 (N = 358).**

| Characteristics | Category | Frequency | % |
|---|---|---|---|
| Place of sleep at night | Street(off street) | 235 | 65.6 |
| | Home (on street) | 123 | 34.4 |
| Street name(for off the street)(n = 235) | Veranda | 145 | 61.7 |
| | Church/Mosque yard | 7 | 3.0 |
| | Bus/railway station | 3 | 1.3 |
| | Empty house or building | 3 | 1.3 |
| | Beside the road | 77 | 32.8 |
| Duration of being on the street | <6month | 54 | 15.1 |
| | 6month-5 year | 287 | 80.2 |
| | >5years | 17 | 4.7 |
| Substance use | Cigarette smoking | 128 | 35.8 |
| | Chat chewing | 38 | 10.6 |
| | Alcohol drinking | 84 | 23.5 |
| | Mastish use | 145 | 40.5 |

## Screening for depression related characteristics of study participants

This study revealed that, the majority (219, 61.2%) of study participants had no depression symptoms for screening. (Fig 3).

## Factors associated with thinness among street adolescents

The Omnibus test of the model coefficient was P value<0.001, which shows that, the full model had a significant prediction. The selected model had a good fit since, the Hosmer-

**Table 3. Illness, health care service, hygiene, and sanitation practices among street adolescents in Adama town, Oromia regional state, Ethiopia, 2023.**

| Characteristics | Category | Frequency | % |
|---|---|---|---|
| Having sickness in the past 3month | No | 150 | 41.9 |
| | Yes | 208 | 58.1 |
| Specify sickness (n = 208) | Respiratory infection | 105 | 50.5 |
| | Diarrhea | 30 | 14.4 |
| | GI infection | 5 | 2.4 |
| | Skin infection | 44 | 21.2 |
| | Headache | 16 | 7.7 |
| | Others | 8 | 3.8 |
| Loss of appetite | No | 232 | 64.8 |
| | Yes | 126 | 35.2 |
| Seek medical care during sickness | No | 216 | 60.3 |
| | Yes | 142 | 39.7 |
| Place of receiving treatment (n = 142) | District hospital | 83 | 58.5 |
| | Referral hospital | 23 | 16.2 |
| | Private hospital | 7 | 4.9 |
| | Pharmacy | 25 | 17.6 |
| | Traditional healer | 4 | 2.8 |
| Taking the prescribed medication properly (n = 142) | No | 40 | 28.2 |
| | Yes | 102 | 71.8 |
| Use latrine | No | 213 | 59.5 |
| | Yes | 145 | 40.5 |
| Body bath per week | No | 237 | 66.2 |
| | Yes | 121 | 33.8 |
| Washing hand before meal | No | 146 | 40.8 |
| | Yes | 212 | 59.2 |
| Source of drinking water | Protected | 279 | 77.9 |
| | Unprotected | 79 | 22.1 |

**Table 4. Dietary intake practice among street adolescents in Adama town, Oromia regional state, Ethiopia, 2023.**

| Characteristics | Category | Frequency | % |
|---|---|---|---|
| How food getting | Leftover from hotels | 297 | 83.0 |
| | Begging | 21 | 5.9 |
| | Cooking | 15 | 4.2 |
| | Buying food | 25 | 7.0 |
| Number of times eat per day | 1 times/day | 57 | 15.9 |
| | 2 times/day | 187 | 52.2 |
| | 3times and morethan /day | 114 | 31.8 |
| Skipped one or more meal /day | No | 86 | 24.0 |
| | Yes | 272 | 76.0 |
| Go sleep hungry | No | 98 | 27.4 |
| | Yes | 260 | 72.6 |
| Dietary diversity score | <5 | 281 | 78.5 |
| | ≥5 | 77 | 21.5 |

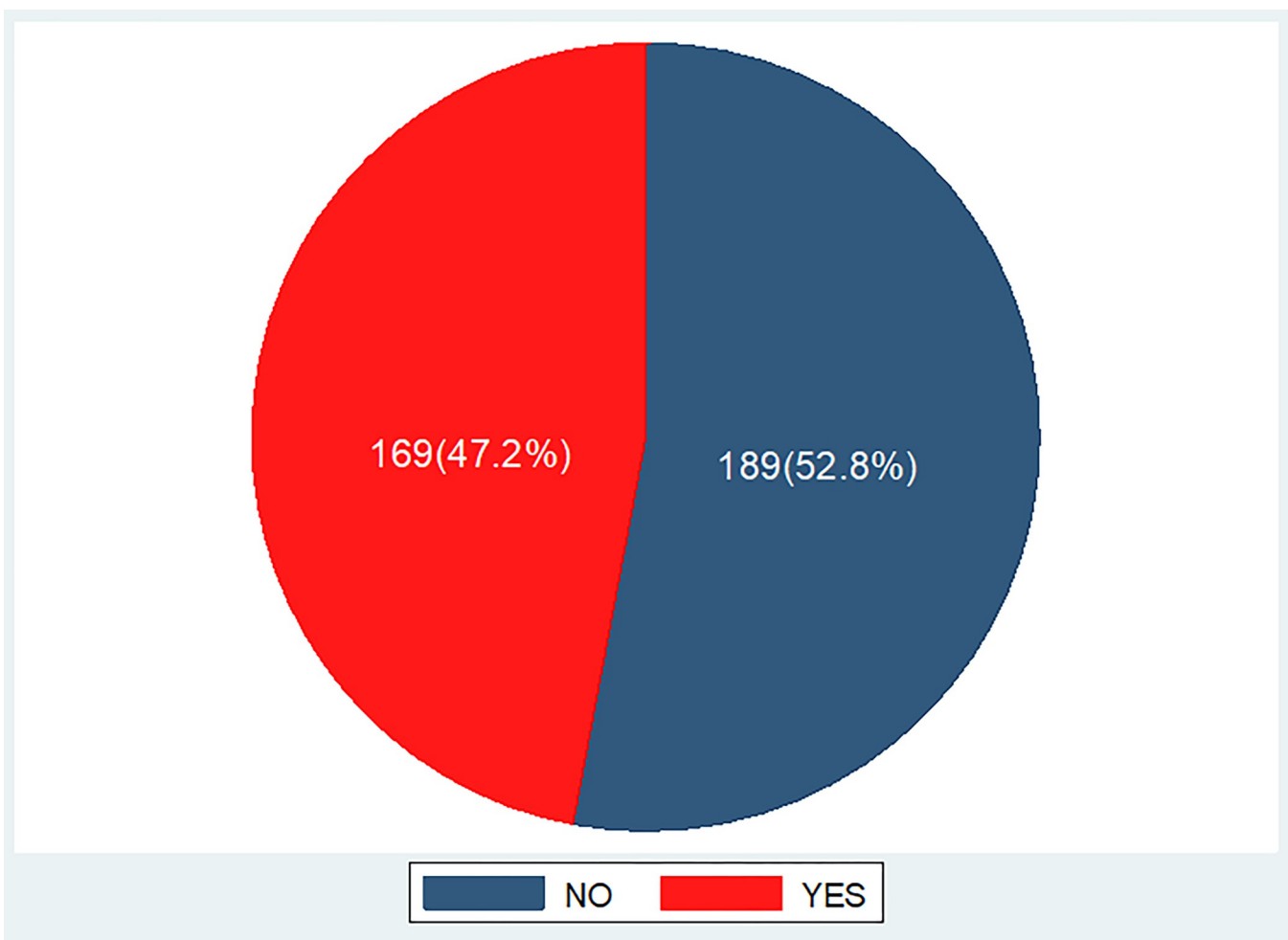

**Fig 1. The overall prevalence of undernutrition among street adolescents in Adama Town, Oromia regional state, Ethiopia, 2023.**

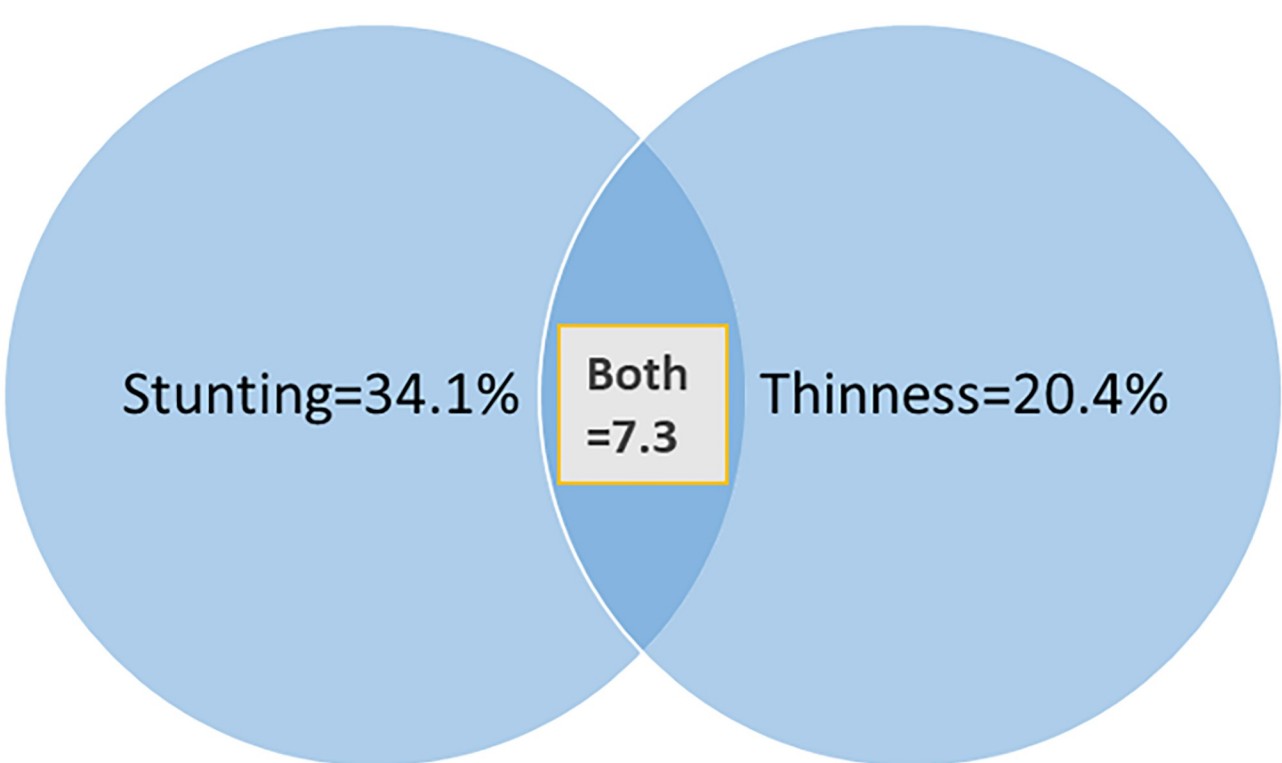

**Fig 2. The prevalence of stunting, thinness and both together among street adolescents in Adama Town, Oromia regional state, Ethiopia, 2023.**

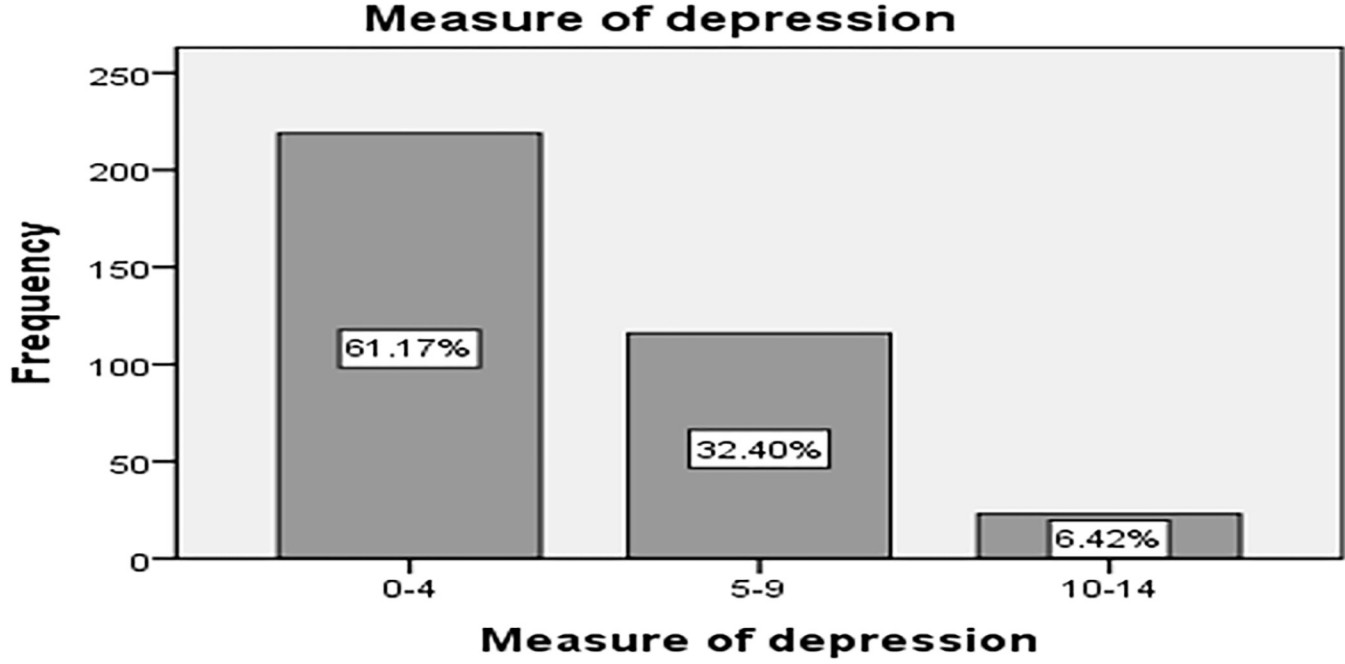

**Fig 3. Percentage of depression among street adolescents in Adama town, Oromia regional state, Ethiopia, 2023.**

Lemeshow test P value was 0.295. Additionally, the model explained 29.4% (Nagelkerke $R^2$) of the thinness. The presence of interaction among independent variables was checked by the multicollinearity test, but there was no significant interaction, as confirmed by the value of the variance inflation factor (VIF), which is less than ten.

In the bivarable regression analysis, factors associated with thinness were age, living with parents, mother's education, smoking cigarettes, drinking alcohol, mastish use, having sickness in the past three months, body bath per week, skipping one or more meals per day, source of drinking water, and screening for depression. However, in multivariable analysis, age, skipping one or more meals per day, using an unprotected water source and using mastish were significantly associated with thinness. A one year increase in the age (AOR = 1.41; 95% CI: 1.17–1.71; P value< 0.001) of the street adolescents was 1.41 times more likely to increase the risk of having thinness. Likewise, street adolescents who skipped one or more meals per day (AOR = 3.50; 95% CI: 1.23–9.94; P value = 0.01) were 3.50 times more likely to have thinness than those who did not skip meals per day. Additionally, street adolescents who had used unprotected water sources (AOR = 3.23; 95% CI: 1.49–6.98; P value = 0.003) were 3.23 times more likely to have thinness than those who had used protected water sources. Moreover, street adolescents who used mastish (AOR = 2.91; 95% CI: 1.19–7.12; P value = 0.01) were 2.91 times more likely to have thinness than those who did not used mastish (Table 5).

**Table 5. Multivariable logistic regression analysis for factors associated with thinness among street adolescents in Adama town, Oromia regional state, Ethiopia, 2023(N = 358).**

| Variable | Category | Thinness | | COR | AOR | P value |
|---|---|---|---|---|---|---|
| | | Yes | No | | | |
| Age | Mean age = 14.92 ±2.35 | | | 1.31(1.16–1.49) | **1.41(1.17–1.71)** * | **.000*** |
| Living with parents | No | 69 | 243 | 1 | 1 | |
| | Yes | 4 | 42 | 0.33(0.11–0.96) | 1.24(0.33–4.57) | 0.74 |
| Mother education | Elementary school | 28 | 80 | 1 | 1 | 0.40 |
| | Secondary school | 6 | 21 | 0.81(0.29–2.22) | 0.88(0.29–2.70) | 0.83 |
| | Higher education | 1 | 6 | 0.47(0.05–4.13) | 0.17(0.01–1.66) | 0.12 |
| | No formal education | 24 | 116 | 0.59(0.32–1.09) | 0.69(0.33–1.43) | 0.32 |
| Smoking cigarete | No | 38 | 192 | 1 | 1 | |
| | Yes | 35 | 93 | 1.90(1.12–3.20) | 0.72(0.29–1.76) | 0.47 |
| Drinking alchol | No | 49 | 225 | 1 | 1 | |
| | Yes | 24 | 60 | 1.83(1.04–3.23) | 0.71(0.27–1.84) | 0.48 |
| Use mastish | No | 31 | 182 | 1 | 1 | |
| | Yes | 42 | 103 | 2.39(1.41–4.04) | **2.91(1.19–7.12)** * | **0.01*** |
| Having sickness | No | 14 | 136 | 1 | 1 | |
| | Yes | 59 | 149 | 3.84(2.05–7.20) | 1.99(0.90–4.42) | 0.08 |
| Body bath per week | No | 56 | 181 | 1 | 1 | |
| | Yes | 17 | 104 | 0.52(0.29–0.95) | 0.69(0.31–1.50) | 0.35 |
| Skipped one or more meals | No | 5 | 81 | 1 | 1 | |
| | Yes | 68 | 204 | 5.40(2.10–13.87) | **3.50(1.23–9.94)*** | **0.01*** |
| Source of drinking water | Protected water source | 49 | 230 | 1 | 1 | |
| | Unprotected water | 24 | 55 | 2.04(1.15–3.62) | **3.23(1.49–6.98)** * | **0.003*** |
| Depression score | 0–4 | 36 | 183 | 1 | 1 | 0.92 |
| | 5–9 | 30 | 86 | 1.77(1.02–3.06) | 0.93(0.44–1.96) | 0.85 |
| | 10–14 | 7 | 16 | 2.22(0.85–5.79) | 0.77(0.21–2.74) | 0.68 |

COR, Crude odds ratio AOR, Adjusted odds ratio.

*Indicates variables that were statistically significant at P value<0.05; 1 = Reference.

## Factors associated with stunting among street adolescents

The model fitness of test have checked in this outcome as well and had a good fit since, the Hosmer- Lemeshow test P value (0.833) and the model explained 23.2% (Nagelkerke $R^2$) of stunting.

In the bivariable analysis, factors associated with stunting were place of sleep at night, duration of being on street, source of income, mastish use, having sickness, loss of appetite, seeking medical care, body bath per week, number of times eating per day, going sleep hungery, skipping one or more meals per day, washing hands before meals and screening for depression. However, in multivariable analysis only skipped one or more meals per day, washing hand before meals and moderate depression were significantly associated with the outcome variable stunting. Street adolescents who had skipped one or more meals per day (AOR = 4.14; 95% CI: 1.87–9.14; P value<0.001) were 4.14 times more likely to have stunting as compared to those who had not skipped the meals. Additionally, those who practiced hand washing before meals (AOR = 0.46; 95% CI: 0.26–0.81; P value 0.008) were 54% less likely to have stunting as compared to those who did not practice hand washing. Moreover, street adolescents who had moderate depression (AOR = 2.93; 95% CI: 1.05–8.15; P value 0.03) were 2.93 times more likely to have stunting than those who did not have depression (Table 6).

## Discussion

This research was aimed to assess the prevalence of undernutrition and its associated factors among street adolescents in Adama, Oromia Regional State, Ethiopia.

The study found that, the prevalence of undernutrition (which includes either thinness or stunting), thinness and stunting was 47.2% (95% CI: 41.9%-52.0%), 20.4% (95% CI: 16.2%-25.1%) and 34.1% (95% CI: 29.3%-39.1%) respectively.

In this study, the prevalence of stunting was 34.1%, which is a concerning issue in Adama town at this moment. This finding is nearly comparable to the study reported in Mysuru (India) (36.8%) and Jimma (Ethiopia) (30.4%) [21,39]. However, this study revealed a greater prevalence than studies conducted in Turkey (12.0%) [40], Benin City, Nigeria (20.4%) [28] and Accra, Ghana (17.7%) [41]. This variation could be due to the difference in sample size and participants' feeding practices. In a study performed in Turkey with a sample size of 75 street children, it was discovered that more than half (56%) of the participants ate three meals a day, and 72% skipped one or more meals. A study in Benin City, Nigeria, also used a sample size of 225 and found that 48.9% had access to eating three times per day. In this study, however, only 33.7% of the study participants ate three times, and 76% skipped one or more meals per day.

This study also revealed a lower prevalence of stunting than studies conducted in Mumbai, India (93% mild to severe) [42], Shabahagn City (61.7%) [43], and Northwest Ethiopia (Bahirdar and Gondar) (46.4%) [18]. This discrepancy may be due to the participants' different age ranges. In those studies, such as in Mumbai, India, the study participants ranged from 6 to 18 years, in Shabahagn city from 6 to > 15 years, and in Northwest, Ethiopia from 5 to 18 years. However, in this study, the participants ranged in age from 10 to 19 years.

This study also found that, the prevalence of being thin was 20.4%. This prevalence value is higher than that in studies conducted in Turkey, Benin and Northwest Ethiopia, which were 2.7%, 13.3% and 15.3%, respectively [18,28,40]. It is likely that the variance can be caused by the participants' varying levels of hygiene and sanitation practices. According to their study, more than half (73%) of participants practiced body bathing, whereas in this study, only 33.8% of participants practiced body bathing.

Studies in Dhaka City (65%), Mumbai (27%), South India (26%) and Jimma, Ethiopia (29.2%), discovered a higher prevalence of thinness than this study [17,21,42,44]. The

**Table 6. Multivariable logistic regression analysis for factors associated with stunting among street adolescents in Adama town, Oromia regional state, Ethiopia, 2023(N = 358).**

| Variable | Category | Stunting | | COR | AOR | P value |
|---|---|---|---|---|---|---|
| | | Yes | No | | | |
| Place of sleep at night | Street (off street) | 86 | 149 | 1 | 1 | |
| | Home (on street) | 36 | 87 | 0.71(0.44–1.14) | 1.59(0.83–3.02) | 0.15 |
| Duration on being on the street | <6month | 12 | 42 | 1 | 1 | 0.19 |
| | 6month-5 years | 106 | 181 | 2.05(1.03–4.06) | 1.72(0.80–3.72) | 0.16 |
| | >5years | 4 | 13 | 1.07(0.29–3.91) | 0.76(0.18–3.16) | 0.70 |
| Source of income | Carrying | 12 | 27 | 1 | 1 | 0.36 |
| | Begging | 62 | 85 | 1.64(0.77–3.49) | 1.73(0.71–4.22) | 0.22 |
| | Shoeshine | 8 | 23 | 0.78(0.27–2.24) | 1.03(0.32–3.32) | 0.95 |
| | Kurale | 7 | 22 | 0.71(0.24–2.12) | 0.71(0.21–2.39) | 0.58 |
| | Daily trading | 1 | 9 | 0.25(0.02–2.20) | 0.23(0.02–2.36) | 0.21 |
| | Petty trading | 9 | 29 | 0.69(0.25–1.91) | 0.71(0.23–2.18) | 0.55 |
| | Garbage picker | 17 | 31 | 1.23(0.50–3.03) | 1.09(0.38–3.07) | 0.86 |
| | Others | 6 | 10 | 1.35(0.39–4.57) | 1.31(0.32–5.29) | 0.70 |
| Use mastish | No | 63 | 150 | 1 | 1 | |
| | Yes | 59 | 86 | 1.63(1.04–2.54) | 1.47(0.83–2.62) | 0.18 |
| Having sickness | No | 37 | 113 | 1 | 1 | |
| | Yes | 85 | 123 | 2.11(1.32–3.35) | 1.43(0.83–2.45) | 0.19 |
| Loss of appetite | No | 71 | 161 | 1 | 1 | |
| | Yes | 51 | 75 | 1.54(0.98–2.42) | 1.27(0.75–2.14) | 0.37 |
| Seek medical care | No | 66 | 150 | 1 | 1 | |
| | Yes | 56 | 86 | 1.48(0.94–2.30) | 1.18(0.71–1.96) | 0.51 |
| Body bath per week | No | 94 | 143 | 1 | 1 | |
| | Yes | 28 | 93 | 0.45(0.27–0.75) | 0.76(0.42–1.39) | 0.38 |
| Number of times eat per day | 1times/day | 27 | 30 | 2.30(1.19–4.46) | 0.93(0.40–2.17) | 0.87 |
| | 2times/day | 63 | 124 | 1.30(0.78–2.16) | 0.77(0.41–1.43) | 0.40 |
| | 3times/day | 32 | 82 | 1 | 1 | 1 |
| Go sleep hungry | No | 28 | 70 | 1 | 1 | |
| | Yes | 94 | 166 | 1.41(0.85–2.34) | 0.52(0.26–1.05) | 0.07 |
| Skipping one or more meals | No | 12 | 74 | 1 | 1 | |
| | Yes | 110 | 162 | 4.18(2.17–8.07) | **4.14(1.87–9.14)*** | **0.00*** |
| Washing hand before meal | No | 69 | 77 | 1 | 1 | |
| | Yes | 53 | 159 | 0.37(0.23–0.58) | **0.46(0.26–0.81)*** | **0.008*** |
| Depression score | 0–4 | 67 | 152 | 1 | 1 | 0.09 |
| | 5–9 | 40 | 76 | 1.19(0.74–1.92) | 0.95(0.55–1.66) | 0.88 |
| | 10–14 | 15 | 8 | 4.25(1.72–10.51) | **2.93(1.05–8.15)*** | **0.03*** |

COR, Crude odd ratio AOR, Ajusted odd ratio.

*Indicates variables that were statistically significant at P-value<0.05; 1 = Reference.

discrepancy might be due to the variation in study participants' disease conditions, hand washing practices and drinking water sources.

In regard to factors that were associated with stunting, this study shown that, skipping one or more meals per day was significantly associated with stunting. Which is inline with the study conducted in Nigeria;where, frequency of meal intake were found to be significantly associated with stunting in a study conducted in Benin, Nigeria [28]. This may be because if meals were skipped, there would be poor dietary intake, which includes insufficient amounts and proportions of energy, protein, vitamins, and minerals, which are linked to macro- and micronutrient deficiencies [2].

Another factor that was significantly associated with stunting was hand washing before meals. Similarly, a study in Rawalpindi found the same association [45]. This may be because handwashing decreases the transmission of disease-causing bacteria, which are caused by poor hygiene and sanitation conditions. Germs, bacteria, or viruses can get into the body through unwashed hands, which causes diarrhea. Repeated episodes of diarrhea cause the loss of micronutrients and prevent food absorption, which results in undernutrition. Improving access to safe water and good hygiene and sanitation practices is important to reduce the incidence of infections and improve nutritional outcomes [2,18,45].

In addition, symptoms of moderate depression were also significantly associated with stunting. Similarly, an analytical review of research in India found a correlation between nutrition and mental health among street children [24]. Although it is not the same population, a study in Kingston, Jamaica, also found a significant relationship between depression and stunting among the non-street population [46]. This may be due to depression, which may influence appetite, food intake, and reduced energy intake, which could lead to improper calorie consumption and impaired nutrient processing in the body and consequently increase the risk of stunting. Street adolescents are more at risk for depression due to environmental factors such as being victimized or bullied, witnessing violence, physical, sexual, or emotional abuse or neglect, and exposure to natural disasters [35].

Some factors, such as age, sex (male), frequency of meals, loss of appetite, substance use, daily duration of being on the streets and source of income were also found to be significantly associated with stunting in other studies, such as in Benin, Rawalpindi, Jimma, and Northwest, Ethiopia; however, they were not in this study [18,21,28,45]. This may be due to sociodemographic compositions, geographical variations, and cultural differences in the study settings.

In regard to factors that were associated with thinness, age was a significantly associated variable, and as age increased, the risk of having thinness also increased. Similarly, a study in the Northwest, Ethiopia, also found a significant association [18]. This may be due to increased physical activity as age increases, combined with poor eating habits, which contribute to increasing the potential risk for older adolescents' thinness. Another justification could be the fact that, older street adolescents are more prone to substance use and have faced different kinds of physical abuse that can increase the chance of having thinness [3].

In addition, thinness was significantly associated with skipping one or more meals per day. A study in Jimma, Ethiopia, also revealed a similar significant association [21]. This could be due to a deficiency in micro and macronutrient intake that are essential for body metabolism and energy supply during the adolescent period. Additionally, a deficiency in sugar or carbohydrates would occur if there were skipped meals for a longer period of fasting. Therefore, the body will manufacture sugar from proteins and fats that are found in muscle and adipose tissue (gluconeogenesis), and when these nutrients are depleted, it results in thinning [5].

Another factor that was significantly associated with thinness was drinking unprotected water. Similarly, studies in the Shabagh area of Dhaka City, Tangail in Bangladesh, and Jimma in Ethiopia found the same association [21,43,47]. This may be because of inadequate access to safe water; contaminated food and water are the main causes of food and waterborne diarrheal disease. Unprotected water is the main transmission route for diseases such as cholera and dysentery [2]. In contrast, a study in Dhaka City, Bangladesh, found no significant association [17]. This may be due to the participants' good hygiene and sanitation practices, of which more than half of them had good hygiene and sanitation practices.

Last, this study also found a significant association between Mastish use and thinness. However, a study in South India did not find a significant association between thinness and substance use in general [44]. The discrepancy may be due to participants' lower number of substance users compared to this study, in their study, participants who used alcohol and

cigarettes were 17% and 24%, respectively, whereas in this study, they were 23.5% and 35.5%, respectively. Although mastishing was not described in their study, in this study, the majority of participants used it. It was significantly associated, perhaps because substance usage has a negative impact on appetite for meals and body composition through decreased intake, nutrient absorption, and dysregulation of hormones that alter the mechanisms of food intake. Mostly street children use mastish to relive hunger and this can lead to undernutrition [18,21].

In other studies, such as Jimma and Northwest, Ethiopaia, factors such as illness, open defication, being female, and a low dietary diversity score were significantly associated with thinness, but they were not statistically significant in this study [18,21]. This may be due to the discrepancy in participants' disease conditions and health service utilization.

## Strength of the study

One of the strengths of this study is that it provides current evidence on factors associated with stunting and thinness among most underprivileged populations. The study included as many important factors as possible that could affect the outcomes.

The other strength of this study is that it added new variables (screening for depression), which was not conducted by other researchers to provide an association with undernutrition among street adolescents.

## Limitations of the study

Using the cross sectional study design by itself is a limitation because it does not help to determine cause and effect relationships.

Micronutrient deficiency is one of the elements of undernutrition but has not been studied in this research due to financial problems and a shortage of time.

As stunting is a chronic undernurition and the prevalence of stunting in this study may not be due to street life; they may have had stunting before they came to the street. Thus, the effect of stunting may not occur within short period of time.

## Conculusion and recommendation

The prevalence of stunting and thinness was a significant public health problem in the study area. Stunting was significantly associated with skipped meals, hand washing, and depression. Where as, thinness was associated with risk factors such as age, skipping meals, drinking unprotected water, and mastish use.

Healthcare providers should emphasized the above modifiable significant factors to tackle undernutrition among street adolescents by collaborating with the Labor and Social affairs office and local nongovernmental organizations. Further research should also be conducted based on specific micronutrient deficiencies to determine the prevalence of each nutrient deficiency.

## Acknowledgments

First, we would like to express our gratitude to Jimma University, Institute of Health, Faculty of Health Sciences, and School of Nursing for approving this paper.

Second, we would also like to thank the Adama Town Administration's Labor and Social Affairs Office for their support.

Last but not least, we would like to give our heartfelt appreciation for data collectors, supervisors and to study participants for their willingness to participate in this study.

## Author Contributions

**Conceptualization:** Tsinukal Tesfaye.

**Data curation:** Tsinukal Tesfaye.

**Formal analysis:** Tsinukal Tesfaye, Fentahun Meseret.

**Funding acquisition:** Tsinukal Tesfaye.

**Investigation:** Tsinukal Tesfaye.

**Methodology:** Tsinukal Tesfaye, Ebissa Bayana Kebede, Vinod Bagilkar, Fentahun Meseret.

**Project administration:** Tsinukal Tesfaye, Fentahun Meseret.

**Supervision:** Tsinukal Tesfaye.

**Validation:** Ebissa Bayana Kebede, Vinod Bagilkar, Fentahun Meseret.

**Visualization:** Ebissa Bayana Kebede, Vinod Bagilkar, Fentahun Meseret.

**Writing – original draft:** Tsinukal Tesfaye, Fentahun Meseret.

**Writing – review & editing:** Ebissa Bayana Kebede, Vinod Bagilkar, Fentahun Meseret.

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
