## [Decision Letter · Decision Letter 0]

7 Nov 2023

PONE-D-23-26853Prevalence of undernutrition and associated factors among street adolescents in adama town, oromia regional state, Ethiopia, 2023: A Cross sectional studyPLOS ONE

Dear Dr. Meseret Wubalem,

Thank you for submitting your manuscript to PLOS ONE. After careful consideration, we feel that it has merit but does not fully meet PLOS ONE’s publication criteria as it currently stands. Therefore, we invite you to submit a revised version of the manuscript that addresses the points raised during the review process. Please submit your revised manuscript by Dec 22 2023 11:59PM. If you will need more time than this to complete your revisions, please reply to this message or contact the journal office at plosone@plos.org. Please include the following items when submitting your revised manuscript:A rebuttal letter that responds to each point raised by the academic editor and reviewer(s). You should upload this letter as a separate file labeled 'Response to Reviewers'.A marked-up copy of your manuscript that highlights changes made to the original version. You should upload this as a separate file labeled 'Revised Manuscript with Track Changes'.An unmarked version of your revised paper without tracked changes. You should upload this as a separate file labeled 'Manuscript'.

We look forward to receiving your revised manuscript.

Kind regards,

David Zadock Munisi, Ph.D

Academic Editor

PLOS ONE

Journal Requirements:

Reviewers' comments:

Reviewer's Responses to Questions

**Comments to the Author**

1. Is the manuscript technically sound, and do the data support the conclusions?

Reviewer #1: Partly

Reviewer #2: Partly

Reviewer #3: Yes

2. Has the statistical analysis been performed appropriately and rigorously? 

Reviewer #1: I Don't Know

Reviewer #2: Yes

Reviewer #3: Yes

3. Have the authors made all data underlying the findings in their manuscript fully available?

Reviewer #1: Yes

Reviewer #2: Yes

Reviewer #3: Yes

4. Is the manuscript presented in an intelligible fashion and written in standard English?

Reviewer #1: No

Reviewer #2: No

Reviewer #3: Yes

5. Review Comments to the Author

Reviewer #1: Thank you for your important research project

Nonetheless, good to respond adequately and clearly to the following inquiries:

a) What is the research gap you wanted to bridge in your study as it is universally understood that most street adolescents in different towns of the country who are disadvantaged both economically and socially are likely to be significantly undermentioned.?

b) What makes your study different from similar studies conducted previously at different towns including Jimma town at Oromia region in Ethiopia?

c) In Adama town as you reported there were over 5000 street adolescents and of them, your sample size 358 street children were identified using convenience sampling method . Do you think that they are adequate enough to represent the source population?

d) In your community based cross sectional study how do you explain the validity and generalizability of your findings obtained from convenience sampling method?.

e) As over 75 % of your study participants were minors, with no guardians or parents in your study setting, how did you solve the challenges of obtaining written consents as well as securing assents to the acceptable level of ethical considerations?

f) In Ethiopian context where by in many instances the living cost is sky rocketing , did you expect most street children to have a good nutritional status; to get daily meals without forcibly skipping ; to have a clean water to drink; to maintain a good personal hygiene ,etc while you are selecting your independent variables ?

g) As you know the street adolescents are very unstable population those move from place to place very frequently, how did you limit their very mobility during your data collection period ?

h) As the main aim of a public health important research is for action or interventions, how do you explain the feasibilities of your recommendations related to street children in today’s Ethiopia?

Reviewer #2: Dear author/s, thank you so much for your works manuscript entitled “Prevalence of undernutrition and associated factors among street adolescents in adama town, oromia regional state, Ethiopia, 2023: A Cross sectional study’.

I have some comments and suggestions;

Abstract

It is not recommended not use abbreviations in abstract.

Please, write key words based on Mesh.

Rewrite the paragraph 2 of introduction with its citations.

Sample size determination

Please, make clear your sample size calculation and write the single population proportion formula.

Why you do not calculate sample size for associated factors?

Sampling procedure and sampling technique is not clear.

By what method you choose kebeles?

Why you choose convenience sampling technique?

Is the calculated sample size proportionally allocated to each selected kebeles?

Did you get list of street adolescents? 5000 adolescents?

Under data collection please collect this sentences “BMI was calculated by weight in kg divided by height squared in cm.”

Did you use different version of SPSS? In abstract it says SPSS V25 and in data processing and analysis is SPSS V26.

Result

How did you 100% response rate?

Please, rewrite result part (remove ambiguous word and redundancy of words).

Reviewer #3: Comments to the Author

This paper addresses a long existing issue of undernutrition in street adolescents. The use of primary data collection methods is notable. The conclusions are consistent with the results but needs more evidence to justify some recommendations made by the authors. The paper needs to be proofread and edited.

Despite the authors’ attempt to address the important issue of undernutrition, there are concerns from introduction to the conclusion section in this study.

Introduction: The introduction should be condensed to a maximum of 2 pages. For instance, paragraphs 2 and 3 have almost the same idea in different paragraphs. Similarly, paragraphs 1 and 4 are discussing the same idea in different paragraphs. So, use connectors and condense them into the same paragraph. Here are also some grammatical problems to be corrected. Unnecessary upper case, but necessary under case. Correct them.

In the last paragraph of the introduction, the author tried to state the knowledge gap as there has not been enough research, particularly in Adama Town. Do you think it is feasible to conduct research in each town in the country? A number of studies have been conducted on street children, specifically on nutrition. Make your study knowledge gap stronger by reviewing the gap in other ways.

Methods: Sample size and Sampling procedures

It is known that Adama City has a sub-city. But you didn’t mention that in your sampling procedure. Clarify the reason.

How did you decide on five kebele? Clarify the procedure. Please describe the procedure using a flow chart.

You used the convenience sampling technique without explaining the reason. Please explain that.

There are streets off and streets on children. You included both. Don't you think that they have different interesting outcomes? How did you manage this?

Your outcome variable operational definition is not clear. You said it was either stunting or thinness. Contrary to this, your dependent variable is stated as the prevalence of undernutrition (stunting and thinness), which needs more clarity.

Result:

The overall prevalence of undernutrition was 169 (47.2%). Include the confidence interval. Even though it is clear how the author measured stunting and wasting, but it is not clear how you measured The overall prevalence of undernutrition Operationalize it objectively.

There is a repetition of the results in the conclusion section of the study. Try to summarize your findings in a different manner than repeating the result.

About figure 1 and 2

The author presented stunting and thinness using a bar graph, which accounted for 122 (34.1%) and 73 (20.4%), respectively, and 26 (7.3%) of both stunting and thinness. A Venn diagram could be useful in presenting this.

6. PLOS authors have the option to publish the peer review history of their article (what does this mean?). If published, this will include your full peer review and any attached files.

Reviewer #1: No

Reviewer #2: No

Reviewer #3: No

---

## [Author Response · Author response to Decision Letter 0]

11 Nov 2023

Dear respected reviewers and editors, I would like to give my deepest appreciation on behave of all authors for your genuine comments and recommendations; all raised questions and comments were valuable to increase the quality of the manuscript; accordingly, we tried to respond for all concerns raised from you as point by point response as it was attached online on the system under the format of the submission. we are awaiting to here a positive response in this regard. regards, Fentahun M.(CA)

---

## [Decision Letter · Decision Letter 1]

15 Dec 2023

Prevalence of undernutrition and associated factors among street adolescents in adama town, oromia regional state, Ethiopia, 2023: A Cross sectional study

PONE-D-23-26853R1

Dear Dr. Wubalem,

We’re pleased to inform you that your manuscript has been judged scientifically suitable for publication and will be formally accepted for publication once it meets all outstanding technical requirements.

Kind regards,

David Zadock Munisi, Ph.D

Academic Editor

PLOS ONE

Additional Editor Comments (optional):

Reviewers' comments:

Reviewer's Responses to Questions

**Comments to the Author**

1. If the authors have adequately addressed your comments raised in a previous round of review and you feel that this manuscript is now acceptable for publication, you may indicate that here to bypass the “Comments to the Author” section, enter your conflict of interest statement in the “Confidential to Editor” section, and submit your "Accept" recommendation.

Reviewer #1: All comments have been addressed

Reviewer #2: All comments have been addressed

2. Is the manuscript technically sound, and do the data support the conclusions?

Reviewer #1: Yes

Reviewer #2: Yes

3. Has the statistical analysis been performed appropriately and rigorously? 

Reviewer #1: I Don't Know

Reviewer #2: Yes

4. Have the authors made all data underlying the findings in their manuscript fully available?

Reviewer #1: Yes

Reviewer #2: Yes

5. Is the manuscript presented in an intelligible fashion and written in standard English?

Reviewer #1: Yes

Reviewer #2: No

6. Review Comments to the Author

Reviewer #1: As the author tried to address most of my previous inquiries adequately, I have no additional comments. Thus, in my opinion, after the required further assessment by the editorial office experts as per the publication guide line of the journal , this article can be published !

Reviewer #2: Dear Author,

Thank you so much for your great work. All comments has been addressed. I would like to say congratulations for your great work.

7. PLOS authors have the option to publish the peer review history of their article (what does this mean?). If published, this will include your full peer review and any attached files.

Reviewer #1: No

Reviewer #2: No
